# Understanding community member and health care professional perspectives on gender-affirming care—A qualitative study

Stephanie Loo[1,2], Anthony N. Almazan[3], Virginia Vedilago[1], Brooke Stott[1], Sari L. Reisner[1,3,4,5], Alex S. Keuroghlian[1,3,6]*

1 The Fenway Institute, Fenway Health, Boston, Massachusetts, United States of America, 2 Department of Health Law, Policy and Management, Boston University School of Public Health, Boston, Massachusetts, United States of America, 3 Harvard Medical School, Boston, Massachusetts, United States of America, 4 Division of Endocrinology, Diabetes, and Hypertension, Brigham and Women's Hospital, Boston, Massachusetts, United States of America, 5 Department of Epidemiology, Harvard T. H. Chan School of Public Health, Boston, Massachusetts, United States of America, 6 Department of Psychiatry, Massachusetts General Hospital, Boston, Massachusetts, United States of America

* akeuroghlian@partners.org

## Abstract

### Background

Transgender and gender diverse (TGD) people experience significant barriers to accessing affirming health services. There is a paucity of literature examining how both community members and health care professionals (HCPs) understand potential causes and solutions for these barriers, particularly in non-urban settings.

### Objective

We present the first systematic examination of perspectives from community members and HCPs regarding barriers to and solutions for promoting access to gender-affirming health care.

### Design

Study activities were conducted through the Plan and Act for Transgender Health (PATH) Project, a health needs assessment of TGD people. Community members in the catchment area were recruited to participate in focus group discussions about access to gender-affirming health care and optimal health service delivery models in March-October 2019. HCPs were recruited to participate in focus group discussions or in-depth interviews about experiences working with TGD clients. Data were analyzed using an inductive grounded theory approach.

### Setting

25 rural counties in Massachusetts, New York, Connecticut, Vermont, and New Hampshire.

**Data Availability Statement:** All relevant data are within the manuscript and its S1–S4 Appendices files.

**Funding:** PATH Study Funder: The Panjandrum Foundation (http://www.panjandrum.org/) Study Author Support: SR: NIH R21MH1181100 from the National Institute of Health (https://www.nih.gov/) AK: U30CS22742 from the Health Resources and Services Administration Bureau of Primary Health Care (https://www.hrsa.gov/) The funders had no role in study design, data collection and analysis, decision to publish, or preparation of the manuscript.

**Competing interests:** The authors have declared that no competing interests exist.

**Abbreviations:** ASL, American Sign Language; CBO, Community-based organization; EHR, Electronic health record; FGD, Focus group discussion; FQHC, Federally qualified health center; HCP, Health care professional; IDI, In-depth interview; PATH, Plan and Act for Transgender Health; TGD, Transgender or gender-diverse.

### Participants

Study participants included 61 adult TGD community members and 23 HCPs working in the catchment area.

### Results

Both community members and HCPs spoke of the need for connectedness and linkages among disparate health system components for gender-affirming health care. Participants expressed this priority through calls for systems-level improvements within existing services (e.g., expanded data collection, expanded mental health services, inclusive and affirming health care environments, and TGD staff). They also expressed the need for expanded TGD community outreach and engagement (e.g., incorporation of a patient feedback process, TGD health navigators, and resource mapping).

### Limitations

Findings specifically reflect the perspectives of community members and HCPs in the rural New England area. Furthermore, the study sample was predominantly White non-Hispanic.

### Conclusion

Interventions to achieve accessible gender-affirming health care must address the diverse perspectives and needs of both community members and HCPs.

## Introduction

Transgender and gender diverse (TGD) people have a gender identity that differs from societal expectations based on their sex assigned at birth. TGD people experience a disproportionate burden of adverse health outcomes; these disparities are likely multifactorial in origin, including due to the deleterious effects of stigma, systemic discrimination, psychiatric pathologization, economic marginalization, and violence victimization [1]. Notably, all of these risks can occur in the context of health care.

TGD people may seek many different types of health care services. These include behavioral health and primary care services that are responsive to the unique needs and experiences of TGD people. TGD people may also seek gender affirmation through treatments including pubertal suppression, gender-affirming hormone therapy, and gender-affirming surgery. Many of the barriers that contribute to gender identity-based health disparities also contribute to negative health care experiences among TGD people. Several studies have demonstrated that TGD people experience high rates of health care discrimination and limited health care access [2–5]. There is therefore a significant need for research that extends our understanding of contemporary barriers to providing accessible and affirming health services for TGD populations. The existing literature on barriers to gender-affirming health care has identified several key contributors: paucity of trained health care professionals (HCPs) [6–9], uncoordinated care [10, 11], unnecessary gatekeeping [11–13], insurance exclusions [11, 14, 15], and electronic health record (EHR) systems that preclude collection of data specific to TGD populations [16–18]. Many of these barriers are amplified in rural areas, and additional challenges have been documented in non-urban settings. The extant literature, largely but not exclusively

based on U.S. communities, has noted several barriers that are more prominent in non-urban regions: hostile social climate, lack of transportation, long waitlists due to more severe short-ages of trained HCPs, and limited internet access that impedes telehealth utilization [19–24]. While there is a growing body of survey-based research investigating perspectives on barriers to gender-affirming care, the majority of surveys have been conducted in urban settings, and most studies do not distinguish between non-urban and urban TGD communities [19]. Fur-thermore, most studies focus on patients and community members, and much less is known about the perspectives of HCPs [25].

This study seeks to systematically examine how community members and HCPs may differ in their understandings of barriers to care, and in their perspectives on potential ways to address such barriers. The dyadic and interpersonal nature of the patient-provider relationship underscores the need to consider each side's perspective separately, and in conversation with each other, given that each is situated in different roles, positions, and levels of power and authority within the broader structure of the healthcare system [26]. Development of interven-tions to mitigate barriers to gender-affirming health care will be contingent on the integration of viewpoints from both community members and HCPs. The establishment of such a dia-logue is crucial, given the longstanding history of impeded communication between TGD patients and HCPs [27]. In this qualitative investigation, we examine community member and HCP perspectives on barriers to, and solutions for, achieving access to gender-affirming health care for TGD communities. The study is also novel in its focus on non-urban TGD communi-ties, an underserved and understudied TGD population.

## Materials and methods

### Data collection

Study activities were conducted through the Plan and Act for Transgender Health (PATH) Project, a needs assessment of the lived experiences, physical and mental health, and unmet health care and social service needs of TGD children, adolescents, and adults. The study was administered in a geographical catchment area that included 25 predominantly rural counties in five U.S. states: Massachusetts, New York, Connecticut, Vermont, and New Hampshire. Qualitative data were collected through focus group discussions (FGDs) and in-depth inter-views (IDIs), between March and October of 2019.

**Focus group discussion recruitment.** Recruitment for the FGDs consisted of social media posts, event participation, and engagement of local community organizations with a specific focus on services for TGD communities in the catchment area. Events included pride festivals and other gatherings, such as luncheons, dances and drop-in groups hosted by local community organizations serving TGD communities. Inclusion criteria for the community member FGDs consisted of identifying as TGD, speaking and reading English or Spanish, being over the age of 18, and living or working in one of the eligible counties in the catchment area. Inclusion criteria for the HCP FGDs consisted of English-speaking individuals over the age of 18, who work in the role of administrator, clinician, or front-line staff, in a healthcare organization already serving or with the capacity to serve TGD clients that is located within the 25 eligible counties. Eligibility screening for FGD participation took place over the phone prior to the group. After determining eligibility, applicants then participated in a verbal informed consent process, which occurred over the phone or in person if participants did not have access to a telephone or if an interpreter was required for obtaining consent.

Community member FGDs were conducted in-person (n = 63) and online (n = 7); the HCP FGDs were exclusively offered online (n = 7). The online FGDs utilized SecureVideo, a HIPAA-compliant Zoom application, to provide flexibility for participants who faced barriers

to transportation or who preferred the greater anonymity of a virtual forum. Semi-structured FGD facilitator guides were created separately for community members and HCPs, including topics on: access to health care, for participating TGD community members; barriers to and facilitators of accessing care, for participating HCPs; and optimal models of health service delivery, for participating TGD community members (see S1 Appendix for interview guides). All FGDs were approximately 90 minutes in length and conducted in either English, Spanish, or American Sign Language (ASL). During the focus group with deaf and hard-of-hearing community members, the ASL interpreters took turns speaking while signing; the session was audio recorded and then professionally transcribed.

**In-depth interview recruitment.** Participating HCPs who engaged in IDIs included administrators, medical and behavioral health clinicians, and front-line staff from health-care organizations. Behavioral health staff include mental health providers such as clinical psychologists, psychiatrists, and licensed social workers. Medical providers included primary care providers, surgeons, pediatricians, midwives, and obstetrician-gynecologists. We also recruited administrative personnel, such as executive health directors, managers in human resources, C-suite executive directors, and department heads in local healthcare service organizations. The inclusion of administrative personnel was intentional, given that an encompassing, TGD-inclusive health care system goes beyond clinical practice. Participating healthcare organizations included private practices, community health centers, federally qualified health centers (FQHCs), and area hospitals. Inclusion criteria included being over the age of 18, speaking and reading English, and engaging in work that serves TGD patients.

Recruitment emails were sent out to area healthcare and social services organizations to identify key staff members who may be willing to participate. Study staff also intentionally recruited TGD providers who are also TGD community members. These providers self-identified as TGD community members and were also able to speak to the experience as TGD employees of interacting with human resources and the hiring process. Interviewees were also asked for referrals to additional HCPs who may be interested in participating in the PATH Project. Participants underwent an in-person informed consent procedure prior to the interview. IDIs with HCPs were conducted in-person and lasted approximately one hour each. Study staff followed an IDI guide consisting of questions about personal experiences working with TGD patients and clients, as well as the perceived effectiveness of each participant's organization in delivering gender-affirming care.

## Data analysis

All FGDs and IDIs were audio recorded by study staff and then sent to a third-party professional transcription service and transcribed verbatim. Following transcription, study staff reviewed all contents and removed any information that may potentially lead to the identification of a participant or outside party. This study was reviewed, approved, and monitored by The Fenway Institute's Institutional Review Board.

Data were analyzed using an inductive grounded theory approach, allowing for major themes to emerge as reported by community member and HCP participants [28]. The study team used the qualitative data analysis software, NVivo 12, to establish an initial codebook and conduct the analysis. The development of the codebook and the coding were conducted by the PATH Project's team of data analysts (SL, VV, BS). The team met regularly to discuss developing themes on a weekly basis. Any discrepancies in coding were reviewed and reconciled within the data analysis team. Finalized key themes resulting from the IDIs and FGDs were then reviewed and discussed with the full study team and content experts (SR, AK).

## Results

A total of 61 adult community members from the geographic catchment area participated in 10 FGDs lasting approximately 90 minutes, with most of the groups averaging 8–10 participants. These FGDs consisted of adult TGD people, including a group of monolingual Spanish speakers and a group of deaf and hard-of-hearing community members that was facilitated through the use of American Sign Language (ASL) interpreters.

Twenty-three HCPs participated in IDIs lasting approximately 60 minutes, and seven additional HCPs participated in one FGD. HCPs were asked to rate their healthcare organization's capacity to serve TGD patients. Additionally, some of the participating HCPs identified themselves as TGD, which allowed them to discuss their personal experiences as TGD community members with their health care employers regarding recruitment, onboarding, and integration into broader work teams (see Tables 1–3).

### Qualitative analysis results

Community members and HCPs were aligned in their prioritization of connectedness and linkages among disparate health system components for affirming and effective TGD health care. Participants expressed this priority with two overarching themes: 1) The desire for systems-level improvements within existing health care services and 2) The desire for expanded TGD community outreach and engagement.

**Table 1. Sociodemographic characteristics of community member focus group discussion participants (9 total focus groups).**

| Total N = 70 | |
|---|---|
| Gender Identity, N (%) | |
| Woman/Trans Woman | 17 (28%) |
| Man/Trans Man | 19 (31%) |
| Non-Binary | 25 (41%) |
| Sex Assigned at Birth, N (%) | |
| Female | 39 (64%) |
| Male | 22 (36%) |
| Age, years, N (%) | |
| 18–24 | 20 (33%) |
| 25–34 | 17 (28%) |
| 35–44 | 4 (7%) |
| 45–54 | 2 (3%) |
| 55+ | 3 (5%) |
| Declined to Answer | 15 (24%) |
| Race, N (%) | |
| Asian/PI | 2 (3%) |
| Black | 3 (5%) |
| Multi-Racial | 7 (11%) |
| White | 40 (66%) |
| Other | 4 (7%) |
| Decline to Answer | 5 (8%) |
| Ethnicity, N (%) | |
| Hispanic, Latina/o/x, or Spanish | 14 (23%) |
| Not Hispanic, Latina/o/x, or Spanish | 47 (77%) |

**Table 2. Roles of health care professionals who participated in in-depth interviews.**

| Total N = 23 | |
|---|---|
| Role/Position, N | |
| Medical Health Care Professional | 12 |
| Mental Health Care Professional | 4 |
| Administrator (medical credentials) | 3 |
| Administrator (no medical credentials) | 4 |

**Theme 1) Systems-level improvements within existing health services, such as expanding data collection options, expanding mental health services, and having inclusive and affirming health care environments that include TGD-identified HCPs and staff.** *1A. Expanding data collection options.* HCPs reported that recent updates in EHRs had generally allowed for patient pronoun documentation. Nevertheless, they explained that some of the recently implemented EHR features are fraught with errors, such as problems with automated organ inventories. HCPs voiced frustration at not being able to directly customize and edit demographic forms in a manner tailored to meet their needs in serving TGD patients.

> *HCP 4: [H]owever, one thing that we typically do to make families more comfortable is we create an alert in the chart for a preferred name and preferred gender. [W]e use that functionality in our [EHR] system to mark someone's chart so that [an] alert pops up. . .so that, starting with the front desk, they can be addressed in a way that makes them more comfortable.*

> *HCP 14: We don't use [the EHR] as it probably could be used for TGD patients just because [gender identity's] not something we put in the EHR. There is the outpatient group, they use [EHR vendor A], and they have the capability to do popups, so they can have a popup anytime they open that patient's chart that says whatever you want it to say. But we use [EHR vendor B], and I have no idea if it even has that capability, because we're not using it.*

Community members voiced wanting to have gender-inclusive forms available for their health care needs. For instance, they reported that availability of expanded sexual orientation and gender identity (SOGI) data options on demographic forms plays a role in being asked about appropriate or inappropriate reproductive health care needs at clinical practices. Community members whose health care organizations offered expanded demographic forms reported feeling validated in being able to enter their gender identity accurately. Conversely, community members who received health care at organizations that did not offer expanded SOGI demographic options reported feeling invalidated in not being able to accurately report their gender identity.

**Table 3. Roles of health care professionals who participated in a focus group discussion (1 focus group discussion).**

| Total N = 7 | |
|---|---|
| Role/Position, N | |
| Mental Health Care Professional | 1 |
| Administrator (no medical credentials) | 4 |
| Community Health Worker/Health Navigator | 2 |

*Community member, FGD E: I get the depo shot at [health center] so that I don't have to deal with menstruation, because that's awful to me. I find the care there really gender affirming, because they ask what your pronouns are and how you identify gender-wise. [W]hen they ask you who you are dating, there's a section for trans woman, or trans man, that you don't have to explain to someone that such a thing exists, and that you're dating this person. [. . .] I've definitely had other [. . .] experiences with reproductive health that have been a lot less gender affirming.*

*Community member, FGD E: [T]he vast majority of medical forms that I fill out when I go to see any doctor only have male or female options. So when I go to those doctors, I let them assume that I'm a female, because I have a uterus, because I don't see myself having another choice besides writing something on the form that there isn't room for, and then getting into a long and uncomfortable conversation. So that's a frustrating issue most of the time.*

Community members and HCPs were aligned in viewing expanded data collection options as a priority, specifically around collecting SOGI data in demographic forms. HCPs emphasized ease-of-use for technology developments, while community members focused on patient-level experiences of being able to report their SOGI status in an affirming and accurate way.

*1B. Additional structural improvement by* expanding mental health services *for TGD people.* Community members were the primary participants who voiced this priority, emphasizing the existing gaps in and needs for culturally tailored mental health care for TGD people.

*Community member, FGD G: "If you're not going to address our mental health, then why open a center?* 'Cause you just gonna be like everything else. The main thing that we see that's lacking across the board - - no matter where you are - - is the mental health component. And how does look, and how can that be catered to [TGD] identities? Because it works in a binary setting, [it] doesn't necessarily work for a nonbinary setting,"*

Further, community members expressed frustration at HCPs over-emphasizing gender-related topics rather than focusing on more pressing community mental health needs as a priority.

*Community member, FGD E: "I'm here for all the rest of my life, so like, can you not try and work through my gender issues? Because I did that."*

Community members wanted to see increased availability of TGD-responsive mental health care professionals well as flexibility in health care delivery options (e.g., telehealth), which would allow access to mental health services for those who live in rural areas. Some community members voiced concern about primary medical clinicians having access to their mental health clinicians, whereas HCPs emphasized the idea of convenience in having an integrated, "one-stop shop." While HCPs also identified the need for mental health care for the TGD community, they focused more on existing approaches and systems for integrated primary medical and mental health care, emphasizing in particular the lack of access to mental health services in rural settings compared to urban health care systems.

*HCP 15: "[O]ur providers here spend quite a bit of time managing the behavioral health needs of their patients regardless of their gender and they need a little bit of support from behavioral health to do the initial assessment. Yes, there - - you know, it's a bipolar patient*

*that needs lithium and then, you know, you prescribe it as the behavioral health physician and then I follow them up and they see you every six months. That doesn't happen in urban centers."*

There was a recurring theme among HCPs that TGD mental health care is outside the scope of their practice, with primary care providers reporting they are unable to address mental health problems within a limited medical visit.

*HCP 17: "Cause I only have 20 minutes with you and I'm not your therapist."*

*1C. Inclusive and affirming health care environments for TGD communities.* HCPs reported that they and their organizations are assessing and taking multiple steps toward building inclusive and affirming health care environments, for instance: proactively hiring TGD staff at their organizations, developing TGD health care best practice checklists for HCPs, and displaying imagery that celebrates TGD communities prominently in their public spaces.

*HCP 11: We certainly want to be perceived and think of ourselves as a gender-affirming -- I'm not sure if I have the terms right, a practice, and accepting. And, so, I know that we have that but how do we present that to the office? I hope we're doing a good job. [. . .] I think that we have pamphlets and things, educational things about that in our waiting rooms. You know, each of our exam rooms, there's a rainbow that represents that you can feel safe in our environment, if you have these kind of issues."*

Community members also voiced support for building inclusive and affirming environments. They did, however, express skepticism about visual depictions and signals as the sole means of fostering TGD inclusivity and affirmation, cautioning against these symbols becoming mere placeholders that fail to translate into meaningfully responsive care for TGD communities. They advocated for systematic training, overseen with verification by their healthcare organization or external experts, rather than relying upon HCPs' self-report of completion. Traditional measures of expertise, such as years of experience, held less value for community members than community feedback, which they believed would be a more accurate reflection of HCPs' cultural responsiveness serving TGD people.

*Community member, FGD E: "You'll have some [providers] who genuinely put checkboxes in what they specialize and have experience in, and you have other people who will fill every single checkbox, because they want more clients. [I]t's unfortunately a useless self-reporting measure."*

*Community member, FGD E: Get actual trans people to rate your business before you advertise yourself as trans competent.*

*1D. Inclusion of TGD-identified HCPs and staff.* In addition, community members and HCPs spoke about having **TGD HCPs and staff**. Health care administrators spoke about the presence of TGD staff as a benefit and reflection of ongoing efforts at their organizations to provide culturally responsive care for TGD patients. Participating HCPs who self-identified as TGD community members during interviews, however, noted an overwhelming sense of exhaustion and burn-out due to becoming the primary resource for TGD care at their health care organizations.

*HCP 15: We feel, based on our recent recruitments, at least at the provider level, we have [TGD] providers that had been recruited, we certainly have, [cisgender] lesbian providers that we have recruited,. . .not sure that there are many [cisgender] male gay providers that we've recruited. [W]hat makes it easy for [town name] is we're kind of known as a comfortable area to be in for a provider who might fall into the LGBT category or categories.*

*Community member 1, FGD H: The Department of Mental Health, I worked with them for two years, [W]hen I was there. . . to help individuals understand about mental health, but it's. . . everything - - drug, substance abuse, alcohol addiction. . . and PTSD. . . I would suggest two people there, a transgender woman and a transgender male. . . I don't think one person can do it all alone.*

*Community member 2, FGD H [in response to community member 1]: Yeah. . .they would burn out real quickly if they had to do that by themselves. They'd get stampeded. (laughs)*

Community members underscored the need for having TGD HCPs and staff as a necessary part of healthcare organizations providing culturally responsive health care that would meet their needs.

*Community Member, FGD E: [F]or whatever you're trying to do with, to interact with the trans community, hire trans people, hire trans people, hire trans people, hire trans people, because we're going to care more.*

**Theme 2) Expanded TGD community outreach and engagement, such as the inclusion of a patient feedback process, TGD-dedicated health navigation services, and effective community resource mapping.** Along with prioritizing systems-level improvements within existing health services as described above, interviewees and discussants voiced the need for new strategies that would allow for more effective outreach to and engagement of TGD communities by healthcare systems.

*2A. Inclusion of a* patient feedback process. One strategy emphasized by participants included having a ***patient feedback process***, whereby TGD community members could provide feedback to HCPs and healthcare organizations to facilitate accountability and adherence to organizational policies and continue improving TGD health care. Both HCPs and community members were favorable toward this idea but had questions about how to implement such a process effectively and confidentially. While HCPs noted that patient feedback would be helpful, they also reported believing that challenges and barriers faced by TGD communities were more societal than related to the healthcare system *per se*. HCPs voiced skepticism about how much quality improvement a patient feedback process would actually provide. Although community members were amenable to the possibility of patient feedback mechanisms, they voiced the need for such feedback to remain anonymous.

*HCP, FGD P: "[The gap is] we have a medical director, but then [. . .] per each site, their clinician isn't under supervision when they're in the room with the person, and there's just no way to guarantee what happens. [S]o when there is a complaint from a client's perspective, it's hard to - - because of the structure of our organization, it can be hard to get accountability or extra training when things like that do happen."*

*2B. Expansion of TGD* health navigators. Both community members and HCPs expressed significant enthusiasm about expanding **TGD health navigator support staff** to improve linkages between TGD communities and health systems. HCPs viewed health navigators as beneficial for psychosocial support, navigating health systems, and easing the burden on clinicians to assist TGD patients in understanding and meeting their health needs within a fractured healthcare system. HCPs who had experience working with existing health navigators reported these navigators were often specifically employed by a particular clinical specialty or clinical practice site. HCPs reported that health navigators often assisted TGD patients with mental health needs even if their official job description did not focus on patients' mental health.

> *HCP 19*: *[Health navigators are] really well-poised to improve care and have perspective on systems improvements that even clinicians sometimes we're just like - - yeah. I think that. . .they're outside the medical system while working inside the medical system [giving them] insights that clinicians might be not quite as likely to prioritize.*

> *HCP 10*: *"[T]hey [primary care providers] try to stay away from the behavioral health actually. Like, that isn't their goal, but it unfortunately that's the reality of our patients."*

Community members echoed HCPs in articulating the need and desire for health navigators as part of a TGD-responsive healthcare system. Community members with direct experience working with health navigators reported positive experiences, including receiving assistance for themselves or their family members in filling out formal paperwork or in booking or coordinating gender-affirming surgical procedures, particularly in anticipation of insurance barriers.

> *Community member, FGD E*: *They [health center] added this person whose role I'm not clear on, but it seems specifically [on] coordinating trans healthcare through the system, and so when I went [there] after some really negative experiences at our old place, a woman there was really great and gave me the [navigator's] card, and [said], "[L]et me know what your insurance is. Oh OK, here are all these providers with your insurance who can do top surgery." And it meant I wasn't having to do all the footwork there. [I]t's the first time I've had [an] experience that was actually accurate and went smoothly.*

Community members offered a conceptualization for the role of health navigators beyond simply providing administrative assistance with scheduling care and navigating health systems. They described health navigators performing an important function as patient advocates who could serve as mediators and facilitators between patients and HCPs. Their ideal TGD health navigator would assist in buffering TGD patients from poor treatment by non-affirming HCPs, as well as deciphering, interpreting, and discussing potential options with TGD patients for next steps in care. Community members also noted that there needs to be a distinction between generic patient navigators and specialized TGD health navigators, who may themselves often be TGD community members.

> *Community member, FGD C*: *Most of the time they do health navigation with patients, like helping you find the letters that you need, or the referrals that you need, but sometimes they do advocacy work where they might talk to you ahead of time. Or, bridge the gap between what the provider's saying, and what they really mean.*

*Community member, FGD G*: They [health navigators] equipped you with tools and gave you information on how to advocate for yourself within these systems, and a lot of times people don't do that. They just deal with one part of your issue or one part of your person. It's not a wraparound service type of thing.

*2C. Effective community* resource mapping. Finally, both HCPs and community members alluded to the importance of effective ***resource mapping***, meaning a collaborative, iterative effort to aggregate evolving TGD health-related resources available for TGD communities. Both community members and HCPs described having "lists" of culturally responsive HCPs that were shared within and across their networks. HCPs underscored the importance of refer-ring patients to specialists that they personally know, particularly for TGD people living in under-resourced non-urban areas.

*HCP 9*: "And out here in [US state], you have to rely on this web, you know? [W]e have to be in contact with each other and talk with each other about what's going on, keep up-to-date about what the web is looking like and how the web is going and (laughs) whether or not any-one needs to intervene [. . .] in the web."

Community members aligned with HCPs in expressing that careful vetting of HCPs is important, utilizing personal and community networks to learn about specific practitioners, or screening practitioner names through online community platforms.

*Community member, FGD E*: If I'm looking for a new anything, I'm going to go search on [online community platform] first. [B]efore that, I remember when I was looking for top sur-geons a bunch of years ago, and there was a map that was created by trans folks around the country of good surgeons for trans procedures. [But] I know that was just a labor of a bunch of trans folks coming together and putting in their effort to this.

However, community members also stressed how much effort was put into advocacy for themselves or family members and how they often felt like they were the ones educating HCPs on culturally responsive care for TGD people. They expressed frustration that they did not per-ceive HCPs making the same proactive effort toward self-education about TGD health issues.

*Community member, FGD D*: "[I]t's hard to put my finger on what [affirming care] means in terms of mental health counseling. Like, they are affirming and yet at the same time I [still] feel the need to funnel information. [L]ike, I'm sending you these articles because I just read them, I think they're really important for you to know about, and I don't get the feeling that they're doing that for me."

## Discussion

This qualitative study investigated and synthesized community member and HCP perspectives on barriers to, and solutions for, achieving access to gender-affirming health care for non-urban TGD communities in the Northeastern U.S. Our findings indicate that barriers to gen-der-affirming care for both community members and HCPs exist at multiple levels. Systems-level improvements within existing health care services are needed to address barriers to care for TGD people. This study brings new perspective to the body of work concerning commu-nity member and HCP perspectives regarding an ideal gender-affirming services within the healthcare system [6–18].

Our study expands on the current literature by considering both the perspectives of community members and HCPs residing in the same geographic area. We noted several distinct differences between our two stakeholder groups. One point of divergence was considering the role of health navigators. The health navigator role has been shown to overcome gaps in knowledge and experience for both HCPs and patients regarding the specific needs of TGD patients as they interact within the healthcare system [29, 30]. HCPs viewed health navigators as beneficial for psychosocial support, navigating health systems, and easing the burden on HCPs to assist TGD patients in understanding and meeting their health needs, findings that align with prior research that identified navigators as central hubs in assisting with appointments and pre-operative tasks [11, 31, 32]. Community members, however, articulated an expanded role for health navigators. In keeping with prior research [33], community members viewed health navigators as patient advocates who take an active part in facilitating patient-HCP interactions. One existing example of health navigators serving in this capacity is the "Trans Buddy," a transgender patient peer advocacy program in which volunteers advocate and provide both logistical and emotional support for TGD patients during their health care visits [31]. The current study findings highlight the importance of health navigation services in gender-affirming care for TGD patients in rural areas, and the need for future development and evaluation of health navigation services in models of care.

Community members also emphasized the need for expanded mental health services, along with the flexibility to have virtual options for care, given many participants' rural places of residence. While HCPs recognized mental health needs for TGD patients, their perspectives were more focused on how behavioral health needs fall outside their scope of practice for primary care providers. In tandem with believing that primary care providers could not address TGD mental health needs, HCPs focused upon building and adding capacity for mental health staff within the healthcare system. This focus maintains separate silos between primary care and mental health needs, which neglects the holistic, affirming care that TGD community members seek. Previous survey findings with TGD community members recommend that HCPs ought to regard mental health integration as part of standard primary care practice [34].

Another point of difference was considering standardized certifications for TGD care. HCPs described visibility of signs and symbols in their healthcare organizations, or employing TGD staff, as indications of culturally responsive care for TGD communities. While both factors fit recommended best practices for affirming TGD health care [35, 36], community members urged caution regarding reliance on tokenizing symbols without substantively affirming TGD care. Community members were also frustrated by the common experience of needing to teach HCPs about TGD health considerations and voiced concern that symbols were used solely to bring in more patients without proper verification to confirm gender-affirming care. Having a standardized certification system, in addition to patient feedback and publicly available reviews of HCPs by TGD community members, are strategies that community members suggested to verify clinical skills and cultural responsiveness of HCPs. Given the rural settings in which many of these participants live, TGD HCPs may be less visible or "out" due to concern for stigma and discrimination [37]. These individuals could experience more professional burnout if they become the sole "expert" on TGD health care at their healthcare organization. Administrators must ensure adequate supports and checks are in place to prevent inadvertently exploiting TGD HCPs. One possible strategy to mitigate this risk involves integrating TGD HCPs within full care teams devoted to TGD care.

Community members and HCPs recommended systems-levels changes to support the provision of gender-affirming care for TGD patients (see Table 4). In making systems-level improvements, healthcare organizations can focus on expanded patient-facing data collection options that include gender-inclusive intake forms and EHR systems, along with establishing a

**Table 4. Summary of findings and recommended best practices.**

| Findings | Recommendations for Best Practices |
|---|---|
| *Systems-level improvements within existing health services* | |
| Expanding data collection options | • Gender-affirming intake forms<br>• Gender-affirming EHR systems<br>• Asking all patients/clients about their gender identity as standard practice |
| Expanding mental health services | • Increased clinical skill and cultural responsiveness for primary care providers working with TGD community members<br>• Increased hiring and availability of gender-affirming mental health professionals<br>• Patient/client consents for information-sharing between integrated health care team members |
| Inclusive and affirming health care environments | • Increased visibility of TGD-affirming symbols, including visibility of flags/icons, pronoun pins, and "safe space" signage in common areas.<br>• All staff attendance at mandatory, recurring training on creating a welcoming environment for TGD patients.<br>• Policies and written procedures ensure a system of accountability and include not only interactions with patients, but also outline expectations for a gender-inclusive workplace culture and care teams inclusive of TGD staff |
| Employing TGD-identified HCPs and staff | • Hiring, retaining, and promoting TGD-identified staff at all levels, including in leadership positions with decision-making power |
| *Expanded TGD community outreach and engagement* | |
| Patient feedback process | • A system is developed for patients/clients to give feedback about an experience, staff member, or the organization<br>• Written policies and procedures to hold staff accountable for following gender-affirming care practices |
| Expanding TGD health navigators | • All patients/clients have access to a knowledgeable health navigator whose services include patient advocacy work with insurance companies and HCPs |
| Resource mapping | • Development of a centralized community resource map, which is monitored and updated regularly<br>• Development of a training program and/or certification that serves to guaranty a higher standard of gender-affirming care practices |

standard practice for asking all patients and clients about gender identity [38]. This work is already ongoing in federally qualified health centers, given the inclusion of SOGI demographic data in Uniform Data System requirements by the Bureau of Primary Health Care [39]. In order to expand TGD mental health services, there needs to be increased clinical skill and cultural responsiveness among primary care providers for working with this population, increased hiring and availability of gender-affirming mental health professionals, and established patient consent processes for information sharing between integrated TGD care team members. All of these factors contribute to inclusive and affirming health care environments, which may be reflected through increased visibility of TGD-affirming symbols, such as the presence of flags/icons, pronoun pins, and "safe space" signage in public spaces. Additionally, healthcare organizations can implement mandatory, recurring all-staff trainings on creating a welcoming environment for TGD patients, and establish policies and written procedures to ensure a system of accountability for interactions with patients and expectations regarding maintenance of a gender-inclusive workplace culture. Organizational leaders ought to take actions toward employing TGD HCPs and staff at all levels, including in leadership positions with decision-making power to support systems-level changes.

In order to enhance TGD community outreach and engagement, healthcare organizations can implement processes for patients and clients to give feedback about an experience, staff member, or systems-level issues at the organization. Additionally, organizations can establish written policies and procedures to hold staff accountable for adhering to gender-affirming care practices. Healthcare organizations can standardize access for all TGD patients to a knowledgeable TGD health navigator whose services include patient advocacy work with a broad range of stakeholders, including HCPs and insurance companies. At the community level, TGD community-based organizations (CBOs) can partner with healthcare organizations to develop a centralized community resource map that is monitored and updated regularly. TGD CBOs can further contribute to the development of training and certification programs that serve to ensure a higher and more consistent standard of gender-affirming care practices at local, regional, or national levels.

## Limitations

There were several limitations to this study. First, our findings reflect the perspectives of community members and HCPs specifically interviewed in non-urban New England, which may limit generalizability. Second, our study sample was predominantly White non-Hispanic, thus we are likely missing key perspectives from non-White HCPs and community members in this study. We did hold one FGD for monolingual Spanish speakers, and one for deaf and hard-of-hearing community members that was facilitated through the use of ASL interpreters, in order to recruit a greater diversity of participants and perspectives.

## Conclusions

Our findings underscore that strategies to achieve affirming health care for TGD people ought to address needs and priorities of both patients and HCPs. These strategies are important at individual, interpersonal, and systems levels in order to achieve improvements in health care practice and experiences for both TGD community members and HCPs. Future implementation of strategies to improve culturally responsive, gender-affirming care for TGD communities will benefit from ongoing evaluation efforts that assess the effectiveness of key care components prioritized based on the perspectives of both community members and HCPs.

## Supporting information

**S1 Appendix. PATH interview guide—Providers.**
(DOCX)

**S2 Appendix. PATH focus group guide—Adults.**
(DOCX)

**S3 Appendix. PATH focus group guide—Healthcare providers.**
(DOCX)

**S4 Appendix. PATH focus group guide—Parents.**
(DOCX)

## Acknowledgments

We thank the health care professionals and community members who participated in this project. We also thank Elliott Blattman, MPH for assistance with data cleaning and coding.

## Author Contributions

**Conceptualization:** Virginia Vedilago, Brooke Stott, Sari L. Reisner, Alex S. Keuroghlian.

**Data curation:** Stephanie Loo, Virginia Vedilago, Brooke Stott.

**Formal analysis:** Stephanie Loo, Virginia Vedilago, Brooke Stott.

**Funding acquisition:** Sari L. Reisner, Alex S. Keuroghlian.

**Investigation:** Brooke Stott, Alex S. Keuroghlian.

**Methodology:** Virginia Vedilago, Brooke Stott.

**Project administration:** Virginia Vedilago, Brooke Stott.

**Supervision:** Sari L. Reisner, Alex S. Keuroghlian.

**Writing – original draft:** Stephanie Loo, Anthony N. Almazan, Virginia Vedilago, Brooke Stott, Sari L. Reisner, Alex S. Keuroghlian.

**Writing – review & editing:** Stephanie Loo, Anthony N. Almazan, Virginia Vedilago, Brooke Stott, Sari L. Reisner, Alex S. Keuroghlian.

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
