## [Decision Letter · Decision Letter 0]

26 Mar 2021

PONE-D-20-30329

Understanding Community Member and Health Care Professional Perspectives on Gender-affirming Care – A Qualitative Study

PLOS ONE

Dear Dr. Loo,

Thank you for submitting your manuscript to PLOS ONE. After careful consideration, we feel that it has merit but does not fully meet PLOS ONE’s publication criteria as it currently stands. Therefore, we invite you to submit a revised version of the manuscript that addresses the points raised during the review process.

The manuscript has been evaluated by two reviewers, and their comments are available below.

The reviewers have raised a number of concerns that need attention, and they request additional information on methodological aspects of the study and the statistical analyses.

Could you please revise the manuscript to carefully address the concerns raised?

We look forward to receiving your revised manuscript.

Kind regards,

Vanessa Carels

Staff Editor

PLOS ONE

Journal Requirements:

1. Please ensure that your manuscript meets PLOS ONE's style requirements, including those for file naming. The PLOS ONE style templates can be found athttps://journals.plos.org/plosone/s/file?id=wjVg/PLOSOne_formatting_sample_main_body.pdf and https://journals.plos.org/plosone/s/file?id=ba62/PLOSOne_formatting_sample_title_authors_affiliations.pdf

Additional Editor Comments (if provided):

Reviewers' comments:

Reviewer's Responses to Questions

**Comments to the Author**

1. Is the manuscript technically sound, and do the data support the conclusions?

Reviewer #1: Yes

Reviewer #2: Yes

2. Has the statistical analysis been performed appropriately and rigorously? 

Reviewer #1: N/A

Reviewer #2: Yes

3. Have the authors made all data underlying the findings in their manuscript fully available?

Reviewer #1: Yes

Reviewer #2: No

4. Is the manuscript presented in an intelligible fashion and written in standard English?

Reviewer #1: Yes

Reviewer #2: Yes

5. Review Comments to the Author

Reviewer #1: 1) Yes, the literature has been largely conducted in the States, but not exclusively, though all of the citations are to US work. I would either say that is the case or correct it (there is the work of Ellis et al in the UK, Riggs et al in AU, Veale et al in NZ etc etc)

2) The data analysis section is far too thin: we need more details about the process

3) Does the table need to say 'woman/trans woman'? Seems redundant and marginalising.

4) The results are somewhat unclear. Do we have 2 themes and within them subthemes marked by the words in bold? Further, they are very generic themes/theme titles: 'improving systems' could be a theme in just about any piece of research on trans health.

5) I dont know that this study is novel as claimed. HCPs and community members have been interviewed separately before. This study just did both in the one study, but not TOGETHER. It wasnt a community consult with everyone in the room, it wasnt a dynamic project where the two groups got to engage with one another and hash our barriers and solutions. I would suggest tempering the language about novelty.

6) Gareth Treharne and colleagues has published a number of papers recently on health navigators.

Reviewer #2: This is a valuable paper given the integration of HCP and trans experiences, however there are some revisions needed as described below.

• The introduction is very short and considering this is not a trans-specific journal you should elaborate on the types of medical care that trans people might seek to access.

• I suggest you look at other qualitative explorations of experiences within health care services and incorporate these into your manuscript, as the extant research does extend beyond US and Canada as you state on p.3:

o Bartholomaeus C, Riggs DW, Sansfaçon AP. Expanding and improving trans affirming care in Australia: experiences with healthcare professionals among transgender young people and their parents. Health Sociology Review. 2020:1-14.

o Strauss P, Lin A, Winter S, Waters Z, Watson V, Wright Toussaint D, et al. Options and realities for trans and gender diverse young people receiving care in Australia’s mental health system: findings from Trans Pathways. Australian & New Zealand Journal of Psychiatry. 2020.

o Strauss P, Winter S, Waters Z, Watson V, Wright Toussaint D, Lin A. Perspectives of trans and gender diverse young people accessing primary care and medical transition services: findings from Trans Pathways. International Journal of Transgender Health. 2021.

• The recruitment information is vague – what does “event participation” refer to? And how were community services engaged?

• Were participants consented over the phone at the time of eligibility assessment?

• What was the scope of healthcare services included? E.g. primary care, surgical, endocrinology?

• The information on the number of focus groups and number of participants are not consistent – in methods the description seems to refer to the number of focus groups, please rephrase this if you are referring to the number of participants and clarify the number of focus groups held.

• Were only HCPs interviewed and no community members? This needs to be explained, and also justified as to why community members weren’t interviewed. The experiences can be so personal and some individuals may not want to share in a FGD.

• Was the focus group with deaf and hard-of-hearing community members videorecorded for transcription?

• Table 1a is the first mention of parents participating – this needs to be better described in methods.

• It’s unclear why in Tables 1b and 1c there are roles with 0 participants – why are these listed? What types of medical care professionals were included? Again, need more specificity here as the perspective of primary care is different to surgical, for example.

• There are a lot of administrators who participated, their unique role should be described briefly in the introduction to justify inclusion.

• What is an EHR? There is no explanation of this acronym (I assume electronic health record?) and it is an American term so should be described.

• The results section would be improved through adding headings separating subthemes, as there are no clear themes to easily identify.

• Are health navigators something that exists in the US or something theoretical from the data? This is unclear to international readers based on the descriptions provided.

• Does “behavioral health staff” refer to mental health? This needs to be clearer in the manuscript.

• Given a third of the community sample were young adults, were there any specific needs here?

• The discussion reiterates the results in many places – the discussion should be more focused on recommendations for improvements and how these experiences can be damaging for trans people to experience.

6. PLOS authors have the option to publish the peer review history of their article (what does this mean?). If published, this will include your full peer review and any attached files.

Reviewer #1: No

Reviewer #2: No

---

## [Author Response · Author response to Decision Letter 0]

7 May 2021

Review Comments to the Author 

  

Reviewer #1:  

1) Yes, the literature has been largely conducted in the States, but not exclusively, though all of the citations are to US work. I would either say that is the case or correct it (there is the work of Ellis et al in the UK, Riggs et al in AU, Veale et al in NZ etc etc) 

We have revised this statement as follows: “The extant literature, largely but not exclusively based on U.S. communities…” We now also cite several additional articles from other countries based the suggestions of both Reviewer 1 and 2. [Page 3, Lines 16-19]

2) The data analysis section is far too thin: we need more details about the process 

We have expanded the description of our analytic process with more details as follows: We applied an inductive grounded theory approach for our analysis. All themes were generated from the data as we coded all focus groups and interviews. Any discrepancies in coding were reviewed and reconciled within the team of data analysts, and finalized themes were reviewed, discussed, and agreed upon with the full study team and content experts. We have now also moved our text detailing the transcription process to this data analysis section. The above description of our analytical process was revised and refined in our data analysis section. [Pages 6, Line 18 – Page 7, Line 4]

3) Does the table need to say 'woman/trans woman'? Seems redundant and marginalising. 

Our categories reflect how our participant sociodemographic data were collected in practice. Regardless of how participants identified, we wanted to insure collection of demographics in an affirming manner. Thus, our survey design combined these categories to allow participants to select their identification without forcing any self-disclosures on the part of participants. 

4) The results are somewhat unclear. Do we have 2 themes and within them subthemes marked by the words in bold? Further, they are very generic themes/theme titles: 'improving systems' could be a theme in just about any piece of research on trans health. 

We have now added our subthemes within each of our two main thematic findings in the results section. This provides the reader with a clearer sense of our main findings and themes that are further detailed in the subsequent text and selected participant quotes. We have now also divided each subtheme into separate headings, as suggested by Reviewer 2. [See Results Section]

5) I dont know that this study is novel as claimed. HCPs and community members have been interviewed separately before. This study just did both in the one study, but not TOGETHER. It wasnt a community consult with everyone in the room, it wasnt a dynamic project where the two groups got to engage with one another and hash our barriers and solutions. I would suggest tempering the language about novelty. 

We have revised the sentence “To our knowledge, no prior studies have systematically examined how community members and HCPs may differ in their understandings of these barriers and challenges to care, or in their perspectives on potential ways to address them.” It now states: “This study sought to systematically examine how community members and HCPs may differ in their understandings of barriers to care, and in their perspectives on potential ways to address such barriers.” [Page 4, Lines 1-2]

We believe that bringing together healthcare provider and community member perspectives regarding rural transgender populations is a significant contribution of our study to the published literature regarding transgender healthcare. We have now also removed mention of novelty from the first sentence of the discussion section. 

6) Gareth Treharne and colleagues has published a number of papers recently on health navigators. 

Thank you for this excellent suggestion. We have now reviewed Dr. Treharne’s literature and added in an additional citation to our discussion section regarding health navigators: “The health navigator role has been shown to overcome gaps in knowledge and experience for both HCPs and patients regarding the specific needs of TGD patients as they interact within the healthcare system.” [Page 17, Lines 22-25]

Reviewer #2: This is a valuable paper given the integration of HCP and trans experiences, however there are some revisions needed as described below. 

• The introduction is very short and considering this is not a trans-specific journal you should elaborate on the types of medical care that trans people might seek to access. 

In our new second paragraph of the Introduction, we have now incorporated a discussion of the different types of medical care TGD people might access. [Page 3, Lines 8-27]

• I suggest you look at other qualitative explorations of experiences within health care services and incorporate these into your manuscript, as the extant research does extend beyond US and Canada as you state on p.3: 

o Bartholomaeus C, Riggs DW, Sansfaçon AP. Expanding and improving trans affirming care in Australia: experiences with healthcare professionals among transgender young people and their parents. Health Sociology Review. 2020:1-14. 

o Strauss P, Lin A, Winter S, Waters Z, Watson V, Wright Toussaint D, et al. Options and realities for trans and gender diverse young people receiving care in Australia’s mental health system: findings from Trans Pathways. Australian & New Zealand Journal of Psychiatry. 2020. 

o Strauss P, Winter S, Waters Z, Watson V, Wright Toussaint D, Lin A. Perspectives of trans and gender diverse young people accessing primary care and medical transition services: findings from Trans Pathways. International Journal of Transgender Health. 2021. 

We have now added all of these suggested citations; thank you very much for providing these references for us to include within our manuscript. They are much appreciated. 

• The recruitment information is vague – what does “event participation” refer to? And how were community services engaged? 

Events included pride festivals and other gatherings, such as luncheons, dances and drop-in groups hosted by local community organizations serving TGD communities. [Page 4, Line 27 – Page 5, Line 1] 

• Were participants consented over the phone at the time of eligibility assessment? 

Eligibility screening for FGD participation took place over the phone prior to the group. After determining eligibility, applicants then participated in a verbal informed consent process, which occurred over the phone or in person if participants did not have access to a telephone or if an interpreter was required for obtaining consent. Eligibility screening for FGD participation took place over the phone prior to the group. [Page 5, Lines 7-10]

• What was the scope of healthcare services included? E.g. primary care, surgical, endocrinology? Medical providers included primary care providers, surgeons, pediatricians, midwives, and OBGYN providers. 

Medical providers included primary care providers, surgeons, pediatricians, midwives, and obstetrician-gynecologists. [Page 5, Lines 26-27]

• The information on the number of focus groups and number of participants are not consistent – in methods the description seems to refer to the number of focus groups, please rephrase this if you are referring to the number of participants and clarify the number of focus groups held. 

Thank you for noting this discrepancy. We excluded one focus group from our analysis as they were parents of TGD youth and their sociodemographic data were not collected. We have therefore now removed the reference to the parent group in Table 1a and revised our total focus group sample size to 61 participants. These corrections have been made in our recruitment and results sections. We have now also added the total number of focus groups held to both Table 1a and Table 1c.

• Were only HCPs interviewed and no community members? This needs to be explained, and also justified as to why community members weren’t interviewed. The experiences can be so personal and some individuals may not want to share in a FGD. 

While only health care providers were interviewed, there were special efforts made to include TGD community members who are also providers. These providers self-identified as community members and were able to speak to the experience as TGD employees of interacting with human resources and the hiring process. [Page 6, Lines 8-17]

• Was the focus group with deaf and hard-of-hearing community members videorecorded for transcription? 

The group was conducted with two certified ASL interpreters. These interpreters took turns signing and speaking. The audio recording was then transcribed by a professional transcription service. [Page 5, Lines 18-21]

• Table 1a is the first mention of parents participating – this needs to be better described in methods. 

Thank you for noticing this, we had incorrectly mentioned the parent focus group here. The data from that focus group were not used in this study. We are in the midst of analyzing those data for a different manuscript focusing on the pediatric transgender population. We have now corrected Table 1a in the manuscript, and relevant sections in the Recruitment and Results. [Table 1a]

• It’s unclear why in Tables 1b and 1c there are roles with 0 participants – why are these listed? What types of medical care professionals were included? Again, need more specificity here as the perspective of primary care is different to surgical, for example. 

We have now removed the categories that had 0 participants in Tables 1b and 1c. We have now added a description of the types of health care professionals who participated in our study to the sections on recruitment. [Page 5, Lines 26-27 & Tables 1b and 1c]

• There are a lot of administrators who participated, their unique role should be described briefly in the introduction to justify inclusion. 

Administrative roles included executive health directors, managers in human resources, C-suite executive directors, and department heads in local healthcare service organizations. The inclusion of administrative personnel is important for our study given that an encompassing, TGD-inclusive health care system goes beyond clinical practice. [Page 5, Line 27 – Page 6, Lines 1-4]

• What is an EHR? There is no explanation of this acronym (I assume electronic health record?) and it is an American term so should be described. 

EHR stands for electronic health record. This acronym was first used in our introduction section at the following passage: “The existing literature on barriers to gender-affirming health care has identified several key contributors: paucity of trained health care professionals (HCPs) [6–9], uncoordinated care [10,11], unnecessary gatekeeping [11–13], insurance exclusions [11,14,15], and electronic health record (EHR) systems that preclude collection of data specific to TGD populations [16–18]”. [Page 3, Lines 16-19] 

• The results section would be improved through adding headings separating subthemes, as there are no clear themes to easily identify. 

We have now added our subthemes within each of our two main thematic findings in the results section. This provides the reader with a clearer sense of our main findings and themes that are further detailed in the subsequent text and selected participant quotes. We have now also divided each subtheme into separate headings, as suggested by the reviewer. [See Results]

• Are health navigators something that exists in the US or something theoretical from the data? This is unclear to international readers based on the descriptions provided. 

To our knowledge, health (or patient) navigator/community health worker support staff exist in non-US settings (e.g., existing literature on navigation services in Australia, Canada, and New Zealand) thus we presumed that readers would understand our use of health navigators within the results and discussion section. 

We now further explain the health navigator role in our Discussion section, paragraph 2: “HCPs viewed health navigators as beneficial for psychosocial support, navigating health systems, and easing the burden on HCPs to assist TGD patients in understanding and meeting their health needs, findings that align with prior research that identified navigators as central hubs in assisting with appointments and pre-operative tasks [11, 31, 32]. Community members, however, articulated an expanded role for health navigators. In keeping with prior research [33], community members viewed health navigators as patient advocates who take an active part in facilitating patient-HCP interactions.” 

• Does “behavioral health staff” refer to mental health? This needs to be clearer in the manuscript. 

Thank you for pointing out this potential point of confusion. Yes, our use of “behavioral health staff” refers to mental health. To avoid confusion, we opted to use mental health explicitly throughout our manuscript with the exception of a participant quote where “behavioral health staff” is referenced. Among the changes made include aligning our text in our Results section, subtheme 1B to reflect the use of mental health over behavioral health. 

• Given a third of the community sample were young adults, were there any specific needs here? 

One of our focus groups consisted of college-aged adults. This group in particular noted the challenges of finding affirming TGD healthcare while transitioning from the pediatric to the primary care setting. This finding was unique to this focus group and thus not included in our overarching findings across FGDs and IDIs. We are separately considering a pediatric/youth-focused manuscript, based on a parent focus group, that would further delve into these themes in our younger cohort. 

• The discussion reiterates the results in many places – the discussion should be more focused on recommendations for improvements and how these experiences can be damaging for trans people to experience. 

Thank you for noting this. Given the breadth of our findings from both community and provider perspectives we thought it pertinent to repeat mention of relevant findings throughout our discussion section. We did remove repetition of results in our first paragraph of the Discussion section in response to the reviewer’s comments. As the reviewer notes, we focused on recommendations for improvements, including our expansive Table 2 which focused on recommendations for best practices in response to each of our individual findings.

---

## [Decision Letter · Decision Letter 1]

21 Jul 2021

Understanding Community Member and Health Care Professional Perspectives on Gender-affirming Care – A Qualitative Study

PONE-D-20-30329R1

Dear Dr. Loo,

We’re pleased to inform you that your manuscript has been judged scientifically suitable for publication and will be formally accepted for publication once it meets all outstanding technical requirements.

Kind regards,

Stefano Federici, Ph.D.

Academic Editor

PLOS ONE

Additional Editor Comments (optional):

Reviewers' comments:

Reviewer's Responses to Questions

**Comments to the Author**

1. If the authors have adequately addressed your comments raised in a previous round of review and you feel that this manuscript is now acceptable for publication, you may indicate that here to bypass the “Comments to the Author” section, enter your conflict of interest statement in the “Confidential to Editor” section, and submit your "Accept" recommendation.

Reviewer #1: All comments have been addressed

2. Is the manuscript technically sound, and do the data support the conclusions?

Reviewer #1: Yes

3. Has the statistical analysis been performed appropriately and rigorously? 

Reviewer #1: N/A

4. Have the authors made all data underlying the findings in their manuscript fully available?

Reviewer #1: No

5. Is the manuscript presented in an intelligible fashion and written in standard English?

Reviewer #1: Yes

6. Review Comments to the Author

Reviewer #1: (No Response)

7. PLOS authors have the option to publish the peer review history of their article (what does this mean?). If published, this will include your full peer review and any attached files.

Reviewer #1: No

---

## [Editor Report · Acceptance letter]

5 Aug 2021

PONE-D-20-30329R1 

Understanding Community Member and Health Care Professional Perspectives on Gender-affirming Care – A Qualitative Study 

Dear Dr. Loo:

I'm pleased to inform you that your manuscript has been deemed suitable for publication in PLOS ONE. Congratulations! Your manuscript is now with our production department. 

Kind regards, 

on behalf of

Prof. Stefano Federici 

Academic Editor

PLOS ONE